**Subject Area:**
microbiology/biochemistry/bioinformatics

MinE, eYFP, fusion proteins, Min system, molecular dynamics simulations

**Author for correspondence:**
Barbara Di Ventura
e-mail: barbara.diventura@biologie.uni-freiburg.de

# C-terminal eYFP fusion impairs *Escherichia coli* MinE function

Navaneethan Palanisamy[1,2,3], Mehmet Ali Öztürk[1,2], Emir Bora Akmeriç[1,2] and Barbara Di Ventura[1,2]

[1]Faculty of Biology, Institute of Biology II, and [2]Centers for Biological Signalling Studies BIOSS and CIBSS, University of Freiburg, Schänzlestr. 1, 79104 Freiburg, Germany
[3]Heidelberg Biosciences International Graduate School (HBIGS), University of Heidelberg, 69120 Heidelberg, Germany

NP, 0000-0003-0369-2316; MAÖ, 0000-0002-0840-1402; BDV, 0000-0002-0247-9989

The *Escherichia coli* Min system plays an important role in the proper placement of the septum ring at mid-cell during cell division. MinE forms a pole-to-pole spatial oscillator with the membrane-bound ATPase MinD, resulting in MinD concentration being the lowest at mid-cell. MinC, the direct inhibitor of the septum initiator protein FtsZ, forms a complex with MinD at the membrane, mirroring its polar gradients. Therefore, MinC-mediated FtsZ inhibition occurs away from mid-cell. Min oscillations are often studied in living cells by time-lapse microscopy using fluorescently labelled Min proteins. Here, we show that, despite permitting oscillations to occur in a range of protein concentrations, the enhanced yellow fluorescent protein (eYFP) C-terminally fused to MinE impairs its function. Combining *in vivo*, *in vitro* and *in silico* approaches, we demonstrate that eYFP compromises the ability of MinE to displace MinC from MinD, to stimulate MinD ATPase activity and to directly bind to the membrane. Moreover, we reveal that MinE-eYFP is prone to aggregation. *In silico* analyses predict that other fluorescent proteins are also likely to compromise several functionalities of MinE, suggesting that the results presented here are not specific to eYFP.

## 1. Introduction

The discovery of the green fluorescent protein (GFP) and its deployment as a fluorescent tag to be fused to proteins of interest has brought a real revolution to molecular biology. For the first time, scientists were able to observe in a time-resolved manner the localization of proteins inside living cells [1]. While snapshots obtained with fixed cells via immunofluorescence are sufficient to reveal if a protein is not homogeneously distributed in the cell, only time-lapse microscopy in living cells can tell if that protein moves around in the same cell. GFP and its spectrally shifted variants [2] are exogenous to the model organisms they are typically expressed into; thus, they do not functionally interfere with endogenous cellular processes [3], unless improperly over-expressed [4]. Moreover, being relatively small (approx. 30 kDa), they do not, in most cases, perturb the localization or function of the protein they are fused to [5,6]. However, this is not always true. Often the problem is caused by the position of GFP within the fusion protein: where an N-terminal fusion may be perturbed, for instance, a C-terminal fusion may be fully functional [7–9]. In some rare cases, GFP is simply too large for the protein of interest, thus impairing the function/localization of the latter even when placed in a location where a smaller tag would be tolerated [10].

Thanks to GFP, it was possible to discover that, in *Escherichia coli* cells, there is a set of proteins that quickly oscillate from one pole to the next throughout the entire cell cycle [11]. These oscillating proteins are MinC, MinD and

MinE, which are encoded by the *minB* operon and together are referred to as the Min system. Its function is to place the cytokinetic ring (Z-ring) in the middle of the cell [11] as well as to facilitate chromosome segregation [12,13]. In the absence of the Min system (so-called Δ*minB* strain), the Z-ring can form anywhere in the cell, leading to the formation of chromosome-less mini-cells [14,15].

The Min proteins are very sensitive to fusions to GFP or its derivatives. MinC-GFP does not complement the mini-cell phenotype of a Δ*minC* strain, while GFP-MinC does [8]. MinD-GFP is dysfunctional (because it does not associate with the membrane anymore), while GFP-MinD complements the mini-cell phenotype of a Δ*minDE* strain when co-expressed with native MinE [11]. MinE-GFP has been contradictorily reported to be either fully functional [16] or to not complement the mini-cell phenotype unless co-expressed with GFP-MinD [17]. This sensitivity to fusions is not surprising, considering the intricate set of interactions and biochemical reactions that must take place in order for the Min system to perform its tasks. MinD is an ATPase belonging to a functionally diverse subgroup of ATPases all having a deviant Walker A motif [18]. It associates with the cytoplasmic membrane via a C-terminal amphipathic helix called the membrane targeting sequence (MTS) [19–21]. When bound to ATP, MinD forms a dimer [21], which is stably associated with the membrane in the absence of MinE [19,22]. In order to place the Z-ring at mid-cell, MinD needs to further bind to MinC, to recruit it to the membrane where MinC antagonizes FtsZ polymerization [23–26]. MinD has also been shown to directly bind FtsZ to correctly position FtsZ and MinC, thus activating MinC inhibitory activity towards FtsZ [27]. To oscillate, and therefore acquire the proper localization to consent Z-ring formation at mid-cell, MinD needs to also bind to MinE, which stimulates MinD ATPase activity and eventually leads to the dissociation of the dimer and the release of MinD from the membrane [28–30]. This MinE-mediated local release of MinD from the membrane, its diffusion in the cytoplasm and its re-association with the membrane in the cell area with the lowest MinE concentration are the necessary events for the pole-to-pole oscillations to occur [31]. Finally, to facilitate chromosome segregation, MinD needs to bind to the DNA, although the binding surface and the precise mechanism are not yet clear in this case [12].

Of the three Min proteins, perhaps the most fascinating is MinE. It is a small protein of 88 amino acids that forms dimers as well as higher-order dynamic structures, typically referred to as E-ring [16,32]. The E-ring delimits the shrinking MinD polar zone moving from the centre towards one pole of the cell at each oscillation round. What distinguishes MinE is the conformational change it undergoes from a 6β-stranded form, where the MinD-binding interface is buried, to a 4β-stranded form, where the MinD-binding interface is exposed [29,30]. MinE contains a cryptic N-terminal MTS [33], which is in equilibrium between two states: it is either bound to (closed form) or unbound from (open form) the 6β-stranded form [29]. In the closed form, the MTS is unable to contact the membrane. Thus, MinE is cytoplasmic and inert. In the open form, the MTS can associate with the membrane and the region of MinE that senses MinD (loop region) is free to do so. Once MinE encounters a MinD dimer on the membrane, either when itself bound to the membrane or when being still in the cytoplasm, a series of

conformational changes occur, which eventually lead to the conversion of a β-sheet (β1) and part of the loop region into the so-called contact helix, which is used by MinE to bind MinD [29]. The form of MinE in which the contact helix is present is the 4β-stranded form, which represents the active protein that stimulates MinD ATPase activity triggering its release from the membrane.

As mentioned above, MinE-GFP was reported to be fully functional in one study [16], but impaired in another [17]. While in most *in vitro* studies MinE is visualized with fluorescent dyes rather than GFP or derivatives [34–36], there are cases in which such fusions are employed [37,38]. Thus, it appears that a consensus is missing as to whether a C-terminal fusion of MinE to GFP or its derivatives impairs the protein function or not. Moreover, even when impairment was reported, no mechanistic explanation was provided. Here, we quantitatively compare MinE and MinE C-terminally fused to the enhanced yellow fluorescent protein (MinE-eYFP) in various *in vivo* and *in vitro* assays. Specifically, we first test the complementation of the mini-cell phenotype of a Δ*minB* strain co-expressing MinE or MinE-eYFP with MinC and MinD from a multi-cistronic construct under arabinose induction. With the purified proteins, we perform liposome co-sedimentation assay with MinD as well as MinD and MinC, and analyse the ability of MinE and MinE-eYFP to displace MinC from MinD and MinD from the membrane. Using coarse-grained (CG) replica-exchange molecular dynamics (REMD) simulations, we predict that the presence of the eYFP renders MinE MTS less accessible, while affecting the dimerization interface and the MinD-binding surface only slightly. The predictions, moreover, point to less accessibility of arginine 21 of MinE, which is involved in the stimulation of MinD ATPase activity [39]. We verify the model predictions by performing liposome co-sedimentation assays, MinD-binding assays and size-exclusion chromatography (SEC). We finally study MinE-eYFP *in vivo* localization at different expression levels and show that it has the tendency to aggregate at higher concentrations. CG REMD simulations of MinE-mCherry and MinE-iLOV suggest that these fusions would also be dysfunctional. Taken together, our results indicate that fusing eYFP C-terminally to MinE impairs its ability to displace MinC from MinD, to directly bind to the membrane and to stimulate MinD ATPase activity, and renders the protein prone to aggregation. The results are likely to be generalizable to other fluorescent proteins beyond eYFP.

# 2. Results

## 2.1. MinE-eYFP does not complement the mini-cell phenotype of a ΔminB strain as well as untagged MinE

To compare the ability of MinE and MinE-eYFP to complement the mini-cell phenotype of a Δ*minB* strain, we first cloned the *minB* operon under the arabinose-inducible pBAD promoter in pBAD33 giving rise to pBAD33[MinCDE] (figure 1*a*). We then introduced the DNA sequence coding for a short flexible linker and the *eyfp* gene downstream of the *minE* gene into pBAD33[MinCDE] giving rise to pBAD33[MinCDE-eYFP] (figure 1*a*). Note that we used a monomeric variant of eYFP, namely the

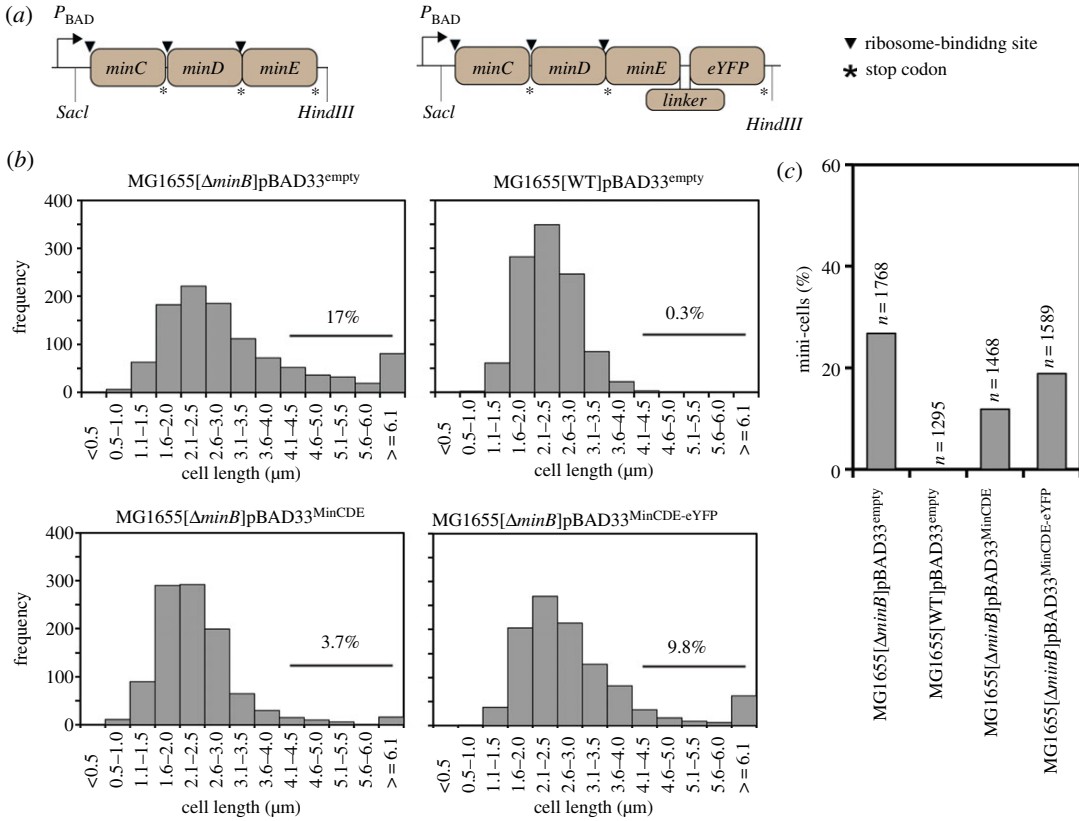

**Figure 1.** MinE-eYFP does not complement the mini-cell phenotype of a Δ*minB* strain as well as untagged MinE. (*a*) Schematics of the constructs. The linker used is 'GSGGG'. (*b*) Cell length distribution of the indicated strain transformed with the indicated plasmid. Expression from the pBAD promoter was induced with 0.0001% arabinose. The number above the black line represents the percentage of cells longer than 4 μm. (*c*) Bar plot showing the percentage of mini-cells for the indicated strains. *n*, total number of cells analysed. The percentage of mini-cells was calculated as the ratio between the number of mini-cells and the total number of cells. (*a,b*) Data from biological triplicates were pooled together and presented.

mutant eYFP^A206K [40,41]. For simplicity, however, we will refer to eYFP^A206K simply as eYFP. We transformed these plasmids as well as empty pBAD33 individually into MG1655Δ*minB* [42]. As a control for normal cell size distribution, we used the wild-type MG1655 strain transformed with empty pBAD33. We performed bright-field microscopy to analyse the cell size distribution as well as the number of mini-cells in the different strains. Under the tested experimental conditions, the *minB* operon expressed from the pBAD33 plasmid did not perfectly complement the mini-cell phenotype of the MG1655Δ*minB* strain (figure 1*b,c*). This is in line with a previous report showing that only integration of a single copy of the *minB* operon into the genome of a Δ*minB* strain allows restoration of the wild-type phenotype [15]. However, here we were interested in the comparison between MinE and MinE-eYFP rather than in a perfect complementation. The histogram shows that the strain expressing MinC, MinD and MinE-eYFP contained approximately 2.6 more longer than wild-type cells than that expressing MinC, MinD and MinE (9.8% versus 3.7%; figure 1*b*). Moreover, the presence of the eYFP on MinE led to 1.6-fold more mini-cells than those obtained with untagged MinE (figure 1*c*).

## 2.2. MinE-eYFP is prone to aggregation

In the complementation experiment described above, we cannot exclude that, despite having used the same arabinose concentration, MinE-eYFP might be expressed at lower levels in the cells than untagged MinE. Moreover, in this assay, it is not possible to separate the contribution of the

individual activities of MinE or MinE-eYFP to the observed phenotype. We therefore decided to move to *in vitro* characterizations of the proteins. In this case, we have full control over the amount of protein, and we can individually study specific biochemical activities using the appropriate assays. To purify MinE and MinE-eYFP, we cloned the respective coding sequences into the pET28a plasmid. The resulting proteins are N-terminally fused to the His (6x) and the T7 tags, separated by a thrombin cleavage site, which can be used to remove the His-tag. When using our standard purification protocol, whereby cells are grown at 37°C, MinE-eYFP was exclusively in the insoluble fraction, while MinE was both in the soluble and insoluble ones (electronic supplementary material, figure S1*a*). Microscopy analysis of *E. coli* Rosetta™ (DE3) pLysS strain expressing MinE-eYFP showed indeed formation of inclusion bodies (electronic supplementary material, figure S1*b*). To investigate whether this behaviour was solely ascribable to the eYFP, we cloned the *eyfp* gene into pET28a and proceeded with the same purification protocol. eYFP was predominantly found in the soluble fraction, suggesting that the fusion of MinE and eYFP is more prone to aggregation than eYFP alone (electronic supplementary material, figure S1*a*). To further evaluate the contribution of eYFP to the aggregation behaviour of MinE-eYFP, we tested the solubility of the fusion between MinE and glutathione *S*-transferase (GST) (MinE-GST), since GST is a commonly used tag for protein purification that is known to aid in protein solubility. Interestingly, MinE-GST was almost exclusively found in the pellet (electronic supplementary material, figure S1*a,c*). However, since both MinE and GST form dimers, we thought this could push the

royalsocietypublishing.org/journal/rsob *Open Biol.* **10**: 200010

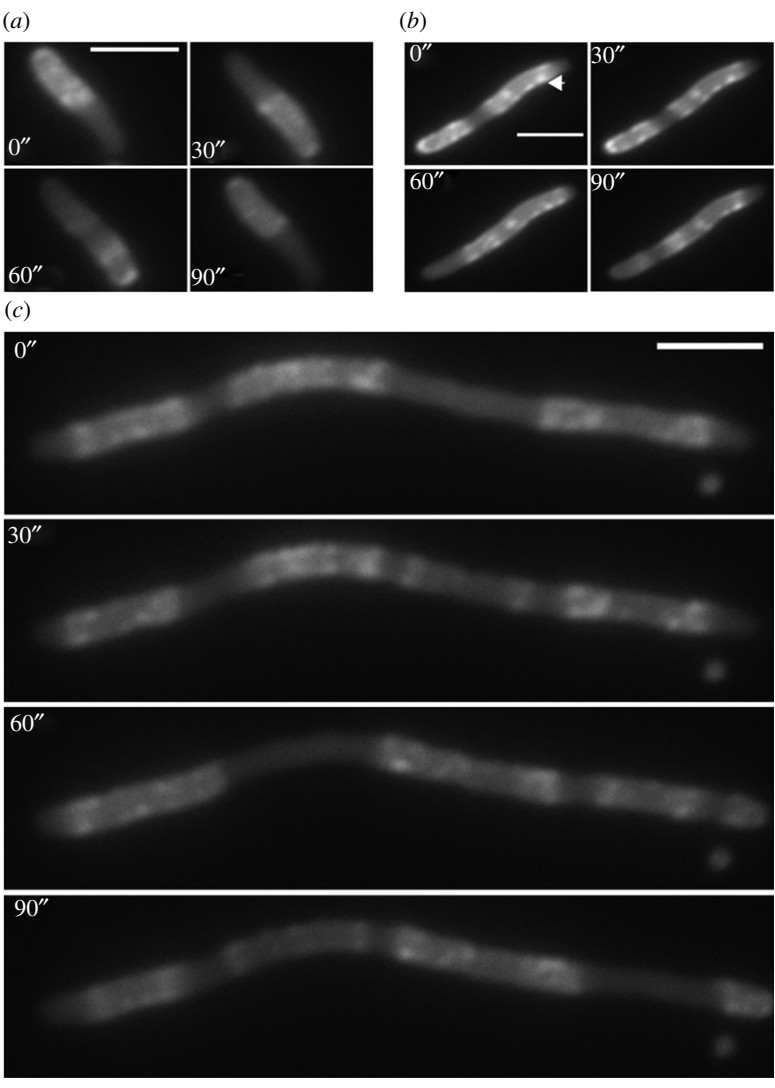

**Figure 2.** Fluorescence time-lapse microscopy analysis of MG1655Δ*minB* cells expressing MinE-eYFP at different concentrations. MG1655Δ*minB* cells were transformed with the MinCDE-eYFP construct shown in figure 1a, and expression from the pBAD promoter was induced with 0.0001% arabinose (a) or 0.1% arabinose (b,c). Scale bar, 3 µm. Arrow, fluorescent cluster that does not move over time.

formation of aggregates. To clarify this point, we further cloned and tested the solubility of MinE-MBP, since the maltose-binding protein (MBP) is monomeric. In this case, we found that the fusion protein was also in the soluble fraction, albeit the insoluble fraction was more abundant (electronic supplementary material, figure S1c). Taken together, the data suggest that MinE C-terminally fused to other proteins has the tendency to aggregate, with the extent of aggregation depending on the specific fusion protein. By lowering the growing temperature to 18°C, we were able to purify MinE-eYFP from the soluble fraction. Given this observation about the tendency of MinE-eYFP to aggregate, we analysed MinE-eYFP localization in living *E. coli* cells at two different expression levels. To this aim, we transformed the pBAD33^MinCDE-eYFP construct into MG1655Δ*minB* cells and induced the expression of the construct with either 0.0001% or 0.1% arabinose. While with 0.0001% arabinose pole-to-pole oscillations and E-rings were visible in 95% of the cells (figure 2a), with 0.1% arabinose, 90% of the cells displayed either fluorescent clusters, which did not move over time (figure 2b), or irregular oscillations (figure 2c). Interestingly, the cells were in this case extremely long (electronic supplementary material, figure S2). This effect was much more pronounced for cells co-expressing MinC and MinD with MinE-eYFP.

## 2.3. MinE-eYFP cannot displace MinC from MinD and MinD from the membrane as well as untagged MinE

To study the ability of MinE and MinE-eYFP to displace MinC from MinD, we performed liposome co-sedimentation assays. In the absence of MinE and in the presence of ATP, MinC is recruited to the liposomes by MinD and is therefore found in the pellet fraction. At increasing MinE concentrations, MinC is increasingly displaced from MinD moving in the supernatant fraction (figure 3a,b). At the same concentrations as MinE, MinE-eYFP was unable to displace the same amount of MinC from MinD. For instance, at 1 µM, MinE displaced 95.6% of MinC from MinD, while MinE-eYFP only 42.4% (figure 3a,c). Using this assay, we cannot investigate whether MinE-eYFP can activate MinD ATPase activity as well as untagged MinE, since the displacement of MinC from MinD is a pre-requisite for this to occur. It could be that MinE-eYFP can activate MinD ATPase activity as well as MinE; however, it cannot displace MinC from MinD and thus MinD remains associated with the liposomes. To clarify this point, we performed liposome co-sedimentation assays with MinD only. We found that MinE-eYFP led to substantially less dissociation of MinD

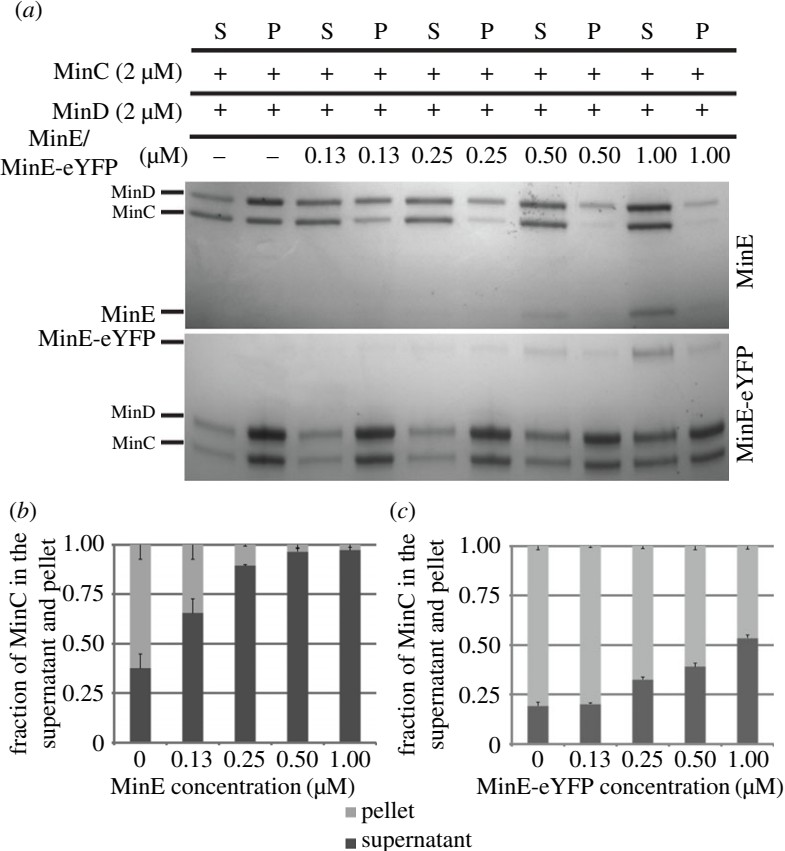

**Figure 3.** MinE-eYFP does not displace MinC from MinD as well as untagged MinE. (*a*) Representative SDS–PAGE analysis of a co-sedimentation assay of MinC and MinD with liposomes in the presence of increasing concentration of MinE or MinE-eYFP. S, supernatant; P, pellet. (*b*,*c*) Quantification of three independent co-sedimentation assays as in (*a*). Values represent mean ± s.e.m.

from the liposomes than that obtained with untagged MinE at all used concentrations (figure 4). For instance, while as little as 0.25 µM MinE were sufficient to release half of the liposome-bound MinD, MinE-eYFP at that concentration did not lead to any release of MinD from the liposomes. The data obtained with this assay could be interpreted as due to a defect of MinE-eYFP in MinD binding, in MinD activation or both.

## 2.4. MinE-eYFP is capable of interacting with MinD

As previously shown, MinD mutated at position D40 and lacking the last 10 amino acids (MinD$^{D40\Delta10}$) can be employed in a pull-down assay to investigate the binding of MinE to MinD [30]. The D40A mutation ensures that ATP hydrolysis does not occur, thus allowing a stable interaction between the proteins. The truncation of the last 10 residues in MinD (Δ10) yields a very soluble protein and does not interfere with the binding between MinD and MinE. Using this assay, we found that MinE-eYFP could pull-down more MinD$^{D40\Delta10}$ than MinE at the used concentration (figure 5*a*,*b*). Interestingly, some binding occurred also in the presence of ADP. Taken together, these results and those of the liposome co-sedimentation assay suggest that MinE-eYFP is specifically impaired in the activation of the ATPase activity of MinD despite being competent to bind to it.

## 2.5. CG REMD simulations indicate that eYFP reduces the accessibility of the MTS and of arginine 21 of MinE

We sought to gain an insight into the molecular mechanism by which eYFP impairs the ability of MinE to stimulate MinD

ATPase activity without affecting its association with MinD, by performing CG REMD simulations. Our aim was to investigate the diffusion accessibility of specific MinE structural elements in the context of the untagged and the fusion proteins. Reduced accessibility of a structural element, such as the MTS, in the fusion protein compared to the untagged one would indicate that this element is somehow 'buried', thus less available for interactions with other surfaces, such as the membrane. Depending on the structural element to analyse, we used either the 6β- or the 4β-stranded form of MinE. Specifically, to investigate whether eYFP would interfere with MinE dimerization, we used the 6β-stranded form; however, we considered the monomer (figures 6*a* and 7*c*). We selected this form because it represents the cytosolic state prior to MinD binding, and we considered the monomer because we asked the question whether dimerization would occur when eYFP is C-terminally fused to MinE. To study the effect of eYFP on membrane and MinD binding as well as on stimulation of MinD ATPase activity, we used the (dimeric) 4β-stranded form (figure 6*b*).

To obtain the full-length 4β- and 6β-stranded structures of *E. coli* MinE, we performed homology modelling using the software MODELLER [43]. As template structures we used the NMR structure of the 6β-stranded form of *Neisseria gonorrhoeae* MinE [44] and the X-ray crystal structure of *E. coli* MinE$^{12–88}$ in its 4β-stranded form lacking the MTS [30]. To capture the dynamic nature of the proteins, we ran CG REMD simulations. Compared with standard MD simulations, REMD simulations provide enhanced sampling of the conformational state of a protein by considering conformations at different temperatures having similar potential

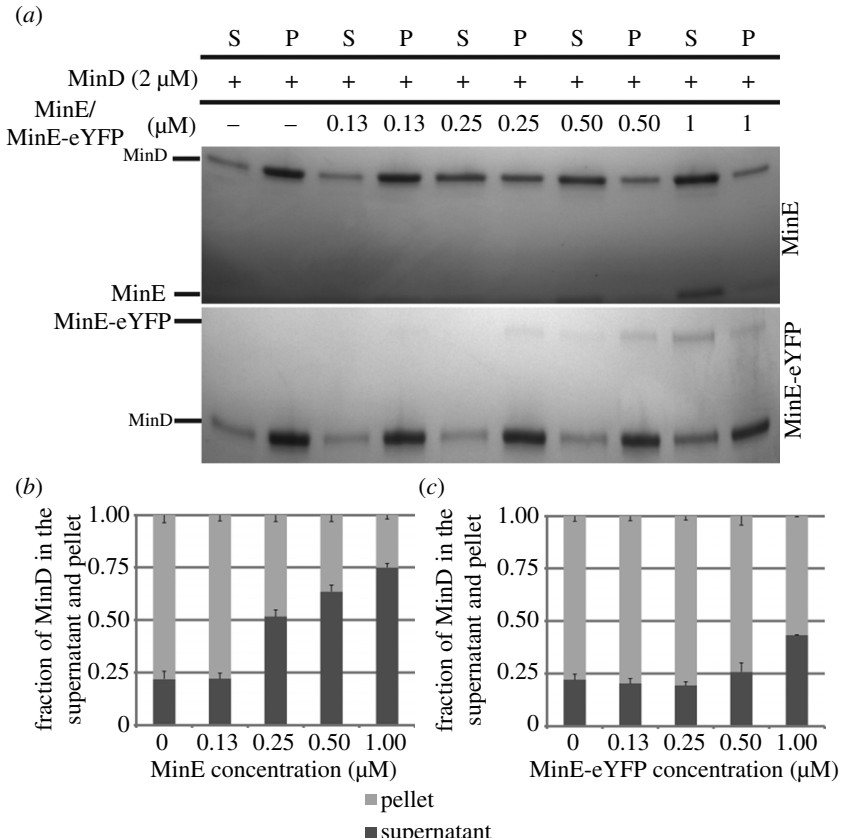

**Figure 4.** MinE-eYFP does not displace MinD from the liposomes as well as untagged MinE. (*a*) Representative SDS–PAGE analysis of a co-sedimentation assay of MinD with liposomes in the presence of increasing concentration of MinE or MinE-eYFP. S, supernatant; P, pellet. (*b,c*) Quantification of three independent co-sedimentation assays as in (*a*). Values represent mean ± s.e.m.

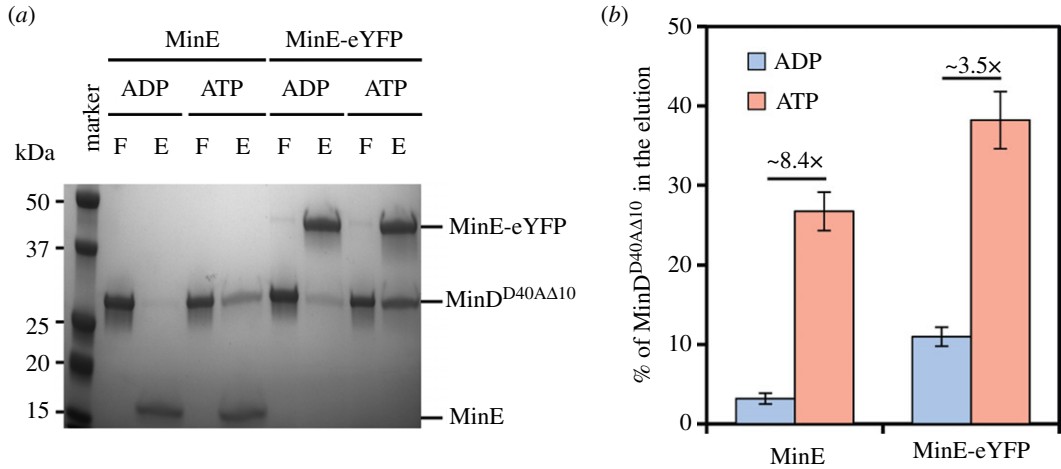

**Figure 5.** MinE-eYFP is capable of interacting with MinD. (*a*) Representative SDS–PAGE analysis of a pull-down assay of MinD$^{D40A\Delta10}$ with MinE or MinE-eYFP in the presence of ADP or ATP. F, flow-through; E, eluate. (*b*) Quantification of three independent pull-down assays as in (*a*). Values represent mean ± s.e.m.

energies [45]. CG atom simplification allows to achieve both accuracy and speed in REMD simulations [45,46].

Since MinE has two dimerization interfaces (figure 6*a*), we analysed the accessibility of both. However, it has to be noted that interface 1, constituted by the β1-sheet, is only transiently used for dimerization, as this part of the protein becomes the contact helix once bound to MinD (figure 6*b*) [29,30]. The CG REMD simulations suggested that the eYFP might render the dimerization interface 1 less accessible, while leaving interface 2 unaffected (figure 7*a*). Given the predominant role of interface 2, we concluded that eYFP was likely not to impair MinE dimerization.

We then analysed the potential effect of eYFP on the direct membrane and MinD binding of MinE by looking at the accessibility of the MTS and the MinD contact helix in the 4β-stranded form of MinE-eYFP, respectively. The CG REMD simulations indicated that the MTS became considerably less accessible, while the MinD interaction surface was only slightly affected (figure 7*b,c*). These theoretical results therefore corroborated our experimental findings that MinE-eYFP binds MinD and offered the prediction that MinE-eYFP might bind less the membrane.

Finally, we investigated the diffusion accessibility of arginine at position 21 (R21) in the 4β-stranded form of

royalsocietypublishing.org/journal/rsob    Open Biol. **10**: 200010

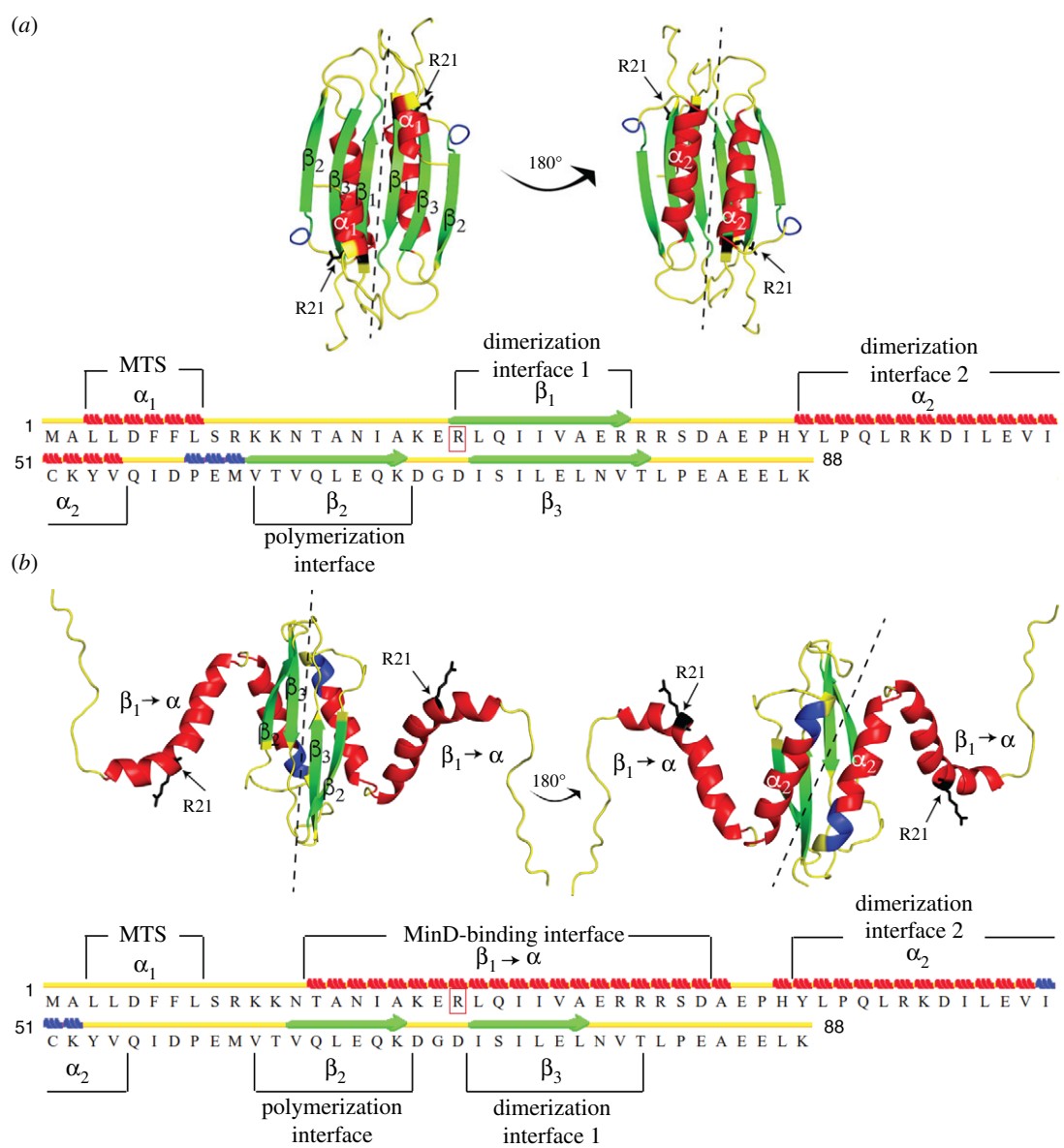

**Figure 6.** Location of important structural features on MinE. (*a,b*) Cartoon representation of the three-dimensional model structures of the 6β- (*a*) and 4β-stranded (*b*) forms of the MinE dimer. Structural elements important for MinE function are shown above and below the amino acid sequence. Secondary structure elements are depicted as follows: red bricks, α-helix; green lines with pointed arrows, β-sheet; blue bricks, 3–10 helix; yellow lines, turn or coil. Structures were depicted in PyMOL (PyMOL Molecular Graphics System, v. 1.8.x, Schrödinger, LLC).

MinE-eYFP. This residue is known to be involved in the stimulation of MinD ATPase activity [39]. R21 is found within the MinD-binding interface (figure 6; electronic supplementary material, figure S3). Interestingly, while overall the MinD-binding interface was only marginally less accessible, the diffusion accessibility score of R21 was substantially lower in MinE-eYFP compared with untagged MinE (figure 7*b,d*). These results suggest that the potential mechanism by which the presence of the eYFP reduces the activation of MinD by MinE is the reduced accessibility of R21.

## 2.6. MinE-eYFP direct membrane association is impaired as computationally predicted

To test the prediction that eYFP reduces the accessibility of MinE MTS, we performed a previously described liposome co-sedimentation assay with only MinE [33]. The cryptic MTS at the N-terminus of MinE is in equilibrium between the closed and the open state. In the open state, the MTS can associate with the membrane, but the association is reversible, and thus, only a minor fraction of MinE is found in the pellet (figure 8*a,b*) compared with, for instance, the amount of MinD that associates with the liposomes in the absence of MinE (figure 4). MinE-eYFP did not associate with the liposomes (figure 8*a,b*); rather, it was found in the pellet regardless of the presence of the liposomes, suggesting aggregation. To understand if the lack of direct membrane association is the cause of the observed impairment of MinE-eYFP in the activation of MinD ATPase activity, we cloned a truncated MinE lacking the first 12 residues constituting the cryptic MTS (MinE[13–88]). This mutant was not yet tested for its ability to remove MinD from liposomes; however, it was shown to be able to activate MinD ATPase activity and to give rise to patterns together with MinD *in vitro* [47]. MinE[13–88] was able to displace MinD from the liposomes as efficiently as the wild-type (figure 8*c,d*), confirming previous observations that the direct association of MinE with the liposomes is not necessary to activate MinD ATPase activity.

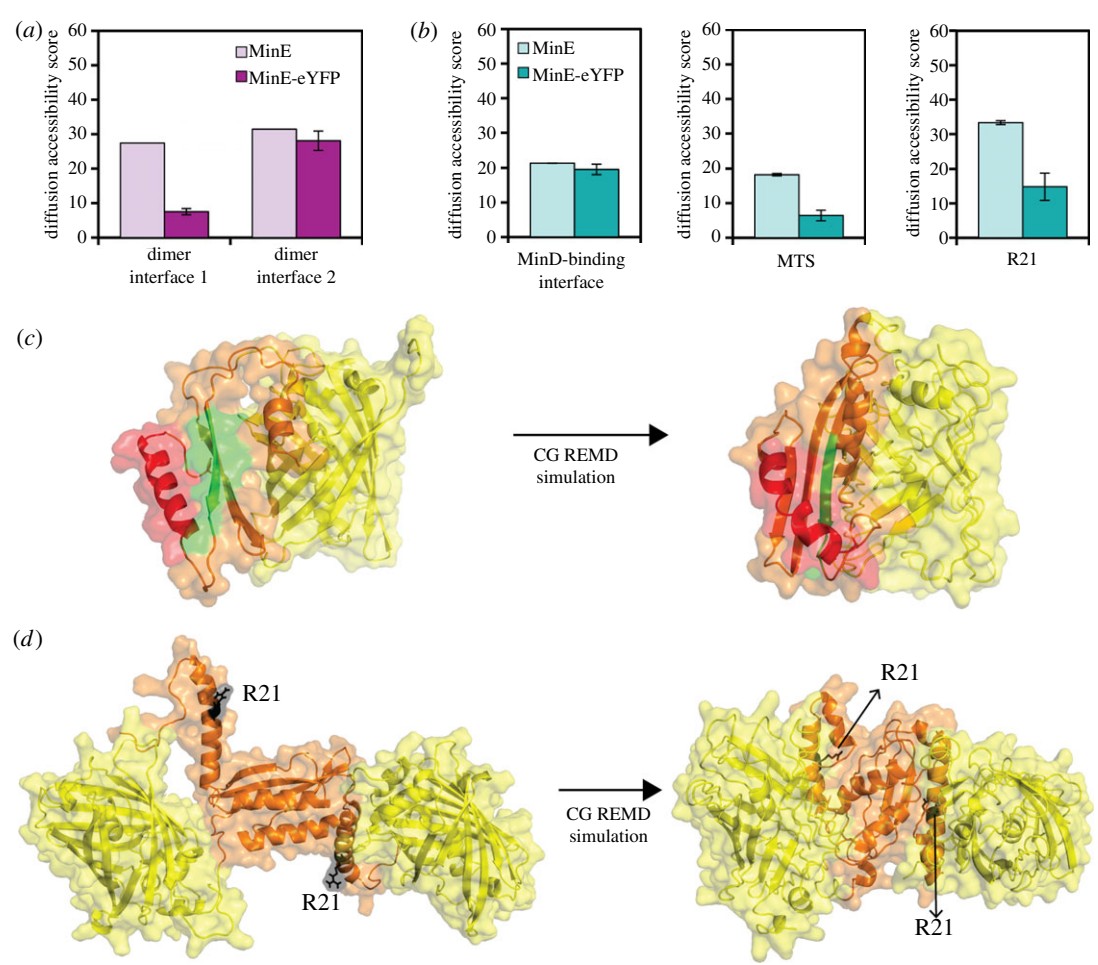

**Figure 7.** Computational analysis of the mechanism by which eYFP impairs MinE function. (*a,b*) Bar plots showing the diffusion accessibility scores for the indicated functional elements. Values represent the mean and error bars represent the standard deviation of five independent simulations. (*c,d*) Cartoon-surface representation of the three-dimensional model structures of the 6β- (*c*) and 4β-stranded (*d*) forms of the MinE-eYFP fusion proteins as CG REMD simulation input and exemplary output. The structure in (*c*) has only three β-sheets as it is the monomer extracted from the 6β-stranded form. eYFP is shown in yellow, MinE in orange, MinE dimer interface 1 in green (*c*), MinE dimer interface 2 in red (*c*) and R21 in black (*d*).

## 2.7. MinE-eYFP dimerization is not affected as computationally predicted

To test the prediction that eYFP does not affect the accessibility of the dimerization interfaces of MinE, we performed SEC with MinE and MinE-eYFP. MinE eluted as a tetramer, in line with previous observations [48], while MinE-eYFP as a dimer (figure 9). As a control, we also performed SEC with eYFP alone and confirmed it to be monomeric (electronic supplementary material, figure S4). These results indicate that MinE-eYFP is indeed not affected in its dimerization.

## 2.8. CG REMD simulations predict that other fluorescent proteins C-terminally fused to MinE are also likely to impair its function

To understand if the results presented here are specific to eYFP or are generalizable, we performed CG REMD simulations with two additional fusions: MinE-mCherry and MinE-iLOV. We selected these two fluorescent proteins because they have a different fold compared to eYFP; additionally, iLOV is very small (approx. 10 kDa); thus, we reckoned it might interfere less with MinE than eYFP. The simulations showed, however, that, despite having no

impact on the MTS or the MinD-binding interface (electronic supplementary material, figure S5*a,b*), these fusions might negatively impact MinE dimerization as well as the ability to activate MinD (electronic supplementary material, figure S5*c,d*).

## 3. Discussion

In this study, we have systematically compared MinE and MinE-eYFP using a combination of *in vitro* and *in vivo* assays, and *in silico* analyses and demonstrated that MinE-eYFP is functionally impaired compared with untagged MinE. Especially the *in vitro* assays allowed us to assess the behaviour of the proteins at the same concentrations, ruling out the possibility that impairment may be due to lower abundance of MinE-eYFP. To gain insights into the potential mechanism by which eYFP may be affecting MinE, we performed CG REMD simulations. These indicated that eYFP decreases the accessibility of MinE MTS as well as of arginine 21, while leaving the dimerization as well as the MinD-binding interfaces unaffected. We validated these predictions by performing liposome co-sedimentation and pull-down assays, and SEC. Since the direct membrane association is not necessary for MinE to dislodge MinD from the liposomes in the co-sedimentation assay used here, as shown in

royalsocietypublishing.org/journal/rsob Open Biol. **10**: 200010

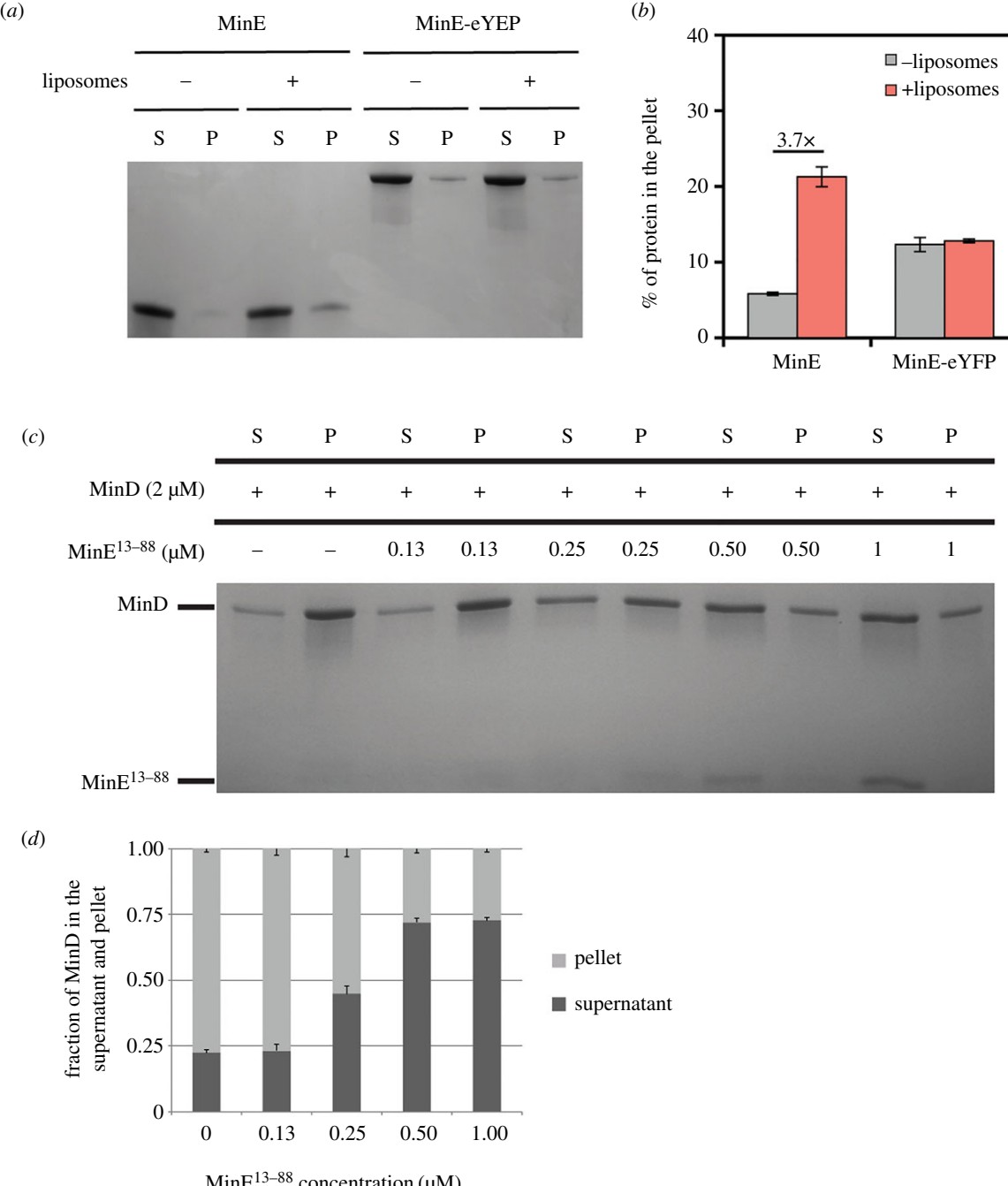

**Figure 8.** The direct membrane binding of MinE-eYFP is impaired. (*a*) Representative SDS–PAGE analysis of a sedimenation assay of MinE or MinE-eYFP with or without liposomes. S, supernatant; P, pellet. (*b*) Quantification of three independent co-sedimentation assays as in (*a*). Values represent mean ± s.e.m. (*c*) Representative SDS–PAGE analysis of a co-sedimentation assay of MinD with liposomes in the presence of increasing concentration of MinE$^{13–88}$. S, supernatant; P, pellet. (*d*) Quantification of three independent co-sedimentation assays as in (*c*). Values represent mean ± s.e.m.

figure 8*c*,*d*, the diminished accessibility of arginine 21 is the likely reason why MinE-eYFP is not as effective as untagged MinE in the activation of MinD ATPase activity. *In vivo*, however, the diminished direct association of MinE with the membrane does affect the Min oscillations and the proper functioning of the Min system [30,33]. Thus, we expect that the interference of eYFP with the association of MinE MTS with the membrane is one of the reasons behind the impairment of MinE-eYFP in the complementation of the mini-cell phenotype (figure 1*b*). It has previously been observed that MinE-GFP oscillates more slowly than GFP-MinD [17]. Our data indicate that the tag leads to functional impairment of MinE, which would justify why the frequency of the oscillations is decreased. However, in order to quantitatively

study the effect of the tag on the frequency of the oscillations, *in vitro* pattern formation should be employed, since this would eliminate the uncertainty of having different protein levels, which may occur *in vivo*.

Interestingly, we found that MinE-eYFP is much less efficient in displacing MinC from MinD compared with untagged MinE (figure 3), despite binding even better to MinD (figure 5). Since the mechanism by which MinE displaces MinC from MinD is not yet clear, we could not computationally analyse the effect of eYFP towards this functionality of MinE. What is known is that MinC and MinE bind MinD at overlapping surfaces, formed upon MinD dimerization [49–53], and that MinE displaces MinC from MinD without requiring the ATP hydrolysis step [54,55].

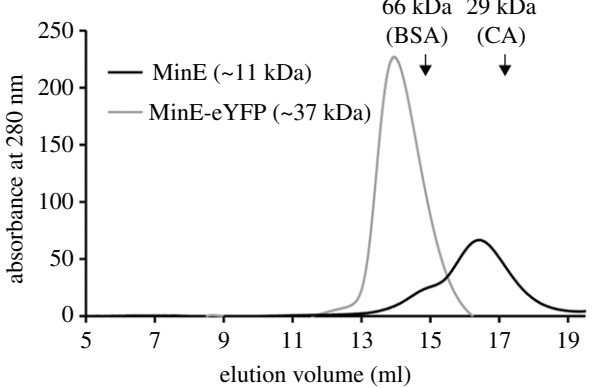

**Figure 9.** MinE-eYFP dimerizes. SEC of MinE and MinE-eYFP, both at 13 µM. The elution positions of the molecular mass standards bovine serum albumin (BSA; 66 kDa) and carbonic anhydrase (CA; 29 kDa) are indicated with arrows.

However, it is not known if MinE and MinC always occupy the same dimeric face or opposing ones, or if both scenarios are possible. In either case, MinE is likely to trigger a conformational change in MinD to displace MinC. Alternatively, MinE would have to directly interact with MinC and then actively repel it from its binding pocket. To our knowledge, no residues have been identified that are involved in these processes. With the current dataset, we cannot discern if eYFP impairs the direct interaction between MinE and MinC and the consequent repulsion of MinC or the conformational change in MinD necessary for the unbinding of MinC.

Finally, we have observed that the fusion to eYFP triggers the aggregation of MinE (electronic supplementary material, figure S1a; figure 8a,b). We believe that this is not due to the intrinsic aggregation propensity of eYFP, since we have shown that eYFP alone is partially soluble (electronic supplementary material, figure S1a) and that MinE tends to aggregate also when fused to highly soluble partners, such as GST and MBP (electronic supplementary material, figure S1a,c). Moreover, we have used the eYFP$^{A206K}$ mutant, which we confirmed to be monomeric (electronic supplementary material, figure S4) [40,41]. We speculate that eYFP favours the open state of MinE by interacting with the MTS; the eYFP would therefore play a similar role as, for instance, the I25R mutation in MinE [30]. However, in case of MinE-eYFP, the MTS is less accessible for membrane insertion exactly due to the interaction with the eYFP contrary to the situation in MinE$^{I25R}$, where the MTS is constitutively bound to the membrane. The open state allows the exposure of the loop region and the β1-sheet that become the contact helix upon MinD sensing [29]. In the absence of MinD or when the MinE concentration exceeds that of MinD, this region could lead to the formation of amyloid-like fibrils as previously reported [56].

In conclusion, given the functional impairment of MinE-eYFP, and the predictions that such impairment might occur in other fusions, such as MinE-mCherry and MinE-iLOV (electronic supplementary material, figure S5), it is advisable to adopt alternative labelling strategies to study properties of the Min system *in vivo*. One such strategy could be click-chemistry-mediated site-specific labelling with fluorescent dyes [57]. This technique is rather demanding, though, as it requires the genetic incorporation of non-canonical amino acids in the protein to be labelled. An alternative is the use of chromobodies, which can be expressed inside living cells and have the advantage to label endogenous molecules [58,59]. However, it remains to be seen if binding of the chromobody to MinE would impair its functions.

# 4. Experimental procedures

## 4.1. Plasmid construction

The *E. coli minC*, *minD* and *minE* genes were individually amplified from the MG1655 genome with primer pairs containing *BamHI* and *HindIII* restriction sites at the 5′ and 3′ ends, respectively. The double-digested DNA fragment was ligated into the multiple cloning site of pET28a and also double-digested with *BamHI* and *HindIII*, yielding plasmids pET28a-MinC, pET28a-MinD and pET28a-MinE. The pET28a-eYFP construct was made similarly, with the *eyfp*$^{a206k}$ gene amplified from plasmid pBDV-15 [12]. The complete pET28a-MinE plasmid, without the stop codon, was amplified by PCR and assembled via the Gibson assembly with the *eyfp* insert to yield pET28a-MinE-eYFP. The pET28-MinE-MBP and pET28-MinE-GST plasmids were also constructed following the same procedure. The *mbp* gene was amplified from the pETM-40 plasmid (kind gift of Gunter Stier, Heidelberg University), while the *gst* gene was ordered as a gene block from IDT (USA). The complete *minB* operon in the negative strand of the *E. coli* MG1655 genome (NCBI accession number: NC_000913.3) starting from chromosomal position 1 224 279 to 1 226 137 was amplified by PCR with a primer pair containing *SacI* and *HindIII* restriction sites at the 5′ and 3′ ends, respectively. The double-digested PCR product was ligated into pBAD33 plasmid and also double-digested with the same restriction enzymes, yielding plasmid pBAD33$^{MinCDE}$. For constructing pET28a-Strep-MinD$^{D40A\Delta10}$ plasmid, first the D40A mutation was introduced into pET28a-MinD by site-directed mutagenesis according to the manufacturer's protocol. Then, the pET28a plasmid backbone was amplified without the region between *NcoI* and *HindIII* restriction sites using pET28a-empty as the template. Using pET28a-MinD$^{D40A}$ as the template, *minD*$^{D40A}$ was amplified by the PCR with a forward primer containing a 5′ Strep-tag sequence and a reverse primer with a stop codon. This reverse primer anneals 33 nucleotides upstream the actual stop codon of *minD*. Finally, the amplified pET28 plasmid backbone and the Strep-MinD$^{D40A\Delta10}$ insert sequence were assembled together by the Gibson Assembly® method. The complete pBAD33$^{MinCDE}$ plasmid, without the stop codon after *minE*, was amplified via PCR resulting in linearized amplicon. The *eyfp* gene was amplified with a forward primer containing the *BamHI* restriction site at the 5′-end followed by nucleotides coding for amino acids 'GGG' and a reverse primer with a stop codon. The backbone and insert were assembled via Gibson Assembly® to yield pBAD33-MinCDE−eYFP.

## 4.2. Microscopy

Bacterial cultures were grown overnight in 4 ml of nutrient broth (LB) with 35 µg ml$^{-1}$ chloramphenicol. The overnight cultures were diluted in fresh tryptone broth (pH adjusted to 7.5) containing chloramphenicol with a starting OD$_{600}$ of

royalsocietypublishing.org/journal/rsob    *Open Biol.* **10**: 200010

0.1. The cultures were grown until $OD_{600}$ of 0.5 after which different concentrations of arabinose were added, and the protein expression was induced for 3 h before performing microscopy. Microscopy slides were prepared by embedding the bacterial samples into 0.5% tryptone agar pads. Imaging was done using a Zeiss Axio-Observer wide-field microscope equipped with a cooled CCD-camera 'AxioCam MRm', an alpha-Plan-APOCHROMAT 100× objective, the Colibri.2 LED light source and the filter set for eYFP. Bright-field images were acquired to measure cell length.

## 4.3. Protein expression and purification

*E. coli* Rosetta™ (DE3) pLysS strain was transformed with the appropriate pET28 plasmid. The overnight bacterial culture was diluted in 1 l fresh nutrient broth containing 50 µg ml$^{-1}$ kanamycin to a starting $OD_{600}$ of 0.1. The culture was grown in a shaker at 37°C until $OD_{600}$ of 0.5 after which 1 mM IPTG was added to induce protein expression. All proteins were induced for 3 h at 37°C, and the cells were harvested by centrifugation. Additionally, MinE-eYFP expression was carried out at 18°C for 12 h to obtain a soluble protein. The pellets were resuspended in lysis buffer (50 mM potassium phosphate pH 8.0, 300 mM NaCl and 10 mM imidazole pH 8.0) containing 0.2 mM ADP, 0.2 mM $MgCl_2$ and cOmplete™ protease inhibitor cocktail tablet (Roche), and lysed by sonication. The lysate was clarified by centrifugation at 20 000 rpm at 4°C and loaded onto an IMAC nickel column (1 ml) using the Bio-Rad NGC chromatography system. The column was washed with wash buffer (same as lysis buffer but with 20 mM imidazole and 10% glycerol) and eluted with elution buffer (same as lysis buffer but the pH of potassium phosphate is 7.5 with further addition of 10% glycerol and 500 mM imidazole). Finally, the elution buffer was replaced with storage buffer (50 mM HEPES-KOH pH 7.25, 150 mM KCl, 10% glycerol and 0.1 mM EDTA pH 8.0) using a P-6 desalting column (10 ml). The purified protein was aliquoted, and the aliquots were stored at −80°C. In all experiments but the pull-down assay, the His-tag was removed from the protein using the Thrombin CleanCleave™ Kit from Sigma-Aldrich.

For the purification of Strep-MinD$^{D40AΔ10}$, Strep-Tactin® cartridge (IBA GmbH, Göttingen, Germany) and the corresponding buffers were used as per the manufacturer's protocol on the Bio-Rad NGC chromatography system. Buffer exchange was performed with the P-6 desalting column (10 ml) as done for the proteins purified with the IMAC nickel column.

## 4.4. Protein solubility analysis

*E. coli* Rosetta™ (DE3) pLysS cells were transformed with the appropriate pET28a plasmid and grown overnight in a nutrient broth containing 50 µg ml$^{-1}$ of kanamycin. The overnight culture was then diluted with 2 ml of fresh nutrient broth containing kanamycin to a starting $OD_{600}$ of 0.1. The culture was grown at 37°C with shaking at 180 rpm. Upon reaching $OD_{600}$ of 0.5, protein expression was induced with 1 mM IPTG, and the culture was grown for additional 3 h. About 1 ml of culture was pelletized and the pellet was resuspended in 150 µl of lysis buffer (see §4.3). The cells were lysed by sonication using a Bioruptor® (Diagenode) with 30 s 'on' and 'off' cycles for 10 min at the highest intensity. The lysate was

cleared by centrifugation at 14 000 rpm for 20 min, and supernatant and pellet were collected separately. The samples were boiled in 1× Laemmli buffer for 12 min and then loaded onto a 12% Mini-PROTEAN® TGX™ precast protein gel. The gel was stained with InstantBlue™ protein stain (Expedeon Ltd, Cambridge, UK) and imaged using the UVP UVsolo touch (Analytik Jena, Germany) gel imaging system.

## 4.5. Liposome co-sedimentation assay

Only MinD (2 µM) or MinD (2 µM) and MinC (2 µM) was/were incubated with 500 µg ml$^{-1}$ liposomes (prepared from *E. coli* phospholipids; kind gift of Chris van der Does, University of Freiburg), 1 mM ATP, 5 mM $MgCl_2$ and different concentrations of MinE or MinE-eYFP for 15 min at room temperature. The reaction mixture was centrifuged at 14 000 r.p.m. for 20 min, and the supernatant and the pellet were collected separately. The samples were then boiled at 95°C in 1× Laemmli buffer and loaded onto a 12% Mini-PROTEAN® TGX™ precast protein gel. The gel was then stained using InstantBlue™ protein stain (Expedeon Ltd, Cambridge, UK) and imaged using UVP UVsolo *touch* (Analytik Jena, Germany).

## 4.6. Pull-down assay

Approximately 5 µM His-MinE or His-MinE-eYFP were incubated with 3 µM Strep-MinD$^{D40AΔ10}$ at room temperature for 15 min in pull-down buffer (50 mM HEPES-KOH pH 7.25, 150 mM KCl, 0.1 mM EDTA pH 8.0, 1 mM ATP, 5 mM $MgCl_2$ and 10% glycerol) with Promega MagneHis™ beads (Promega, USA). Bound proteins were eluted with elution buffer (50 mM Tris pH 7.0 and 500 mM imidazole pH 8.0). The flow-through and elution fractions were boiled at 95°C in 1× Laemmli buffer and loaded onto a 12% Mini-PROTEAN® TGX™ precast protein gel. The gel was then stained using the InstantBlue™ protein stain (Expedeon Ltd, Cambridge, UK) and imaged using the UVP UVsolo *touch* (Analytik Jena, Germany).

## 4.7. Mine membrane-binding assay

About 6 µM MinE or MinE-eYFP were incubated with liposomes (1 mg ml$^{-1}$) in binding buffer (20 mM Tris–Cl pH 7.5, 200 mM sucrose) at 30°C for 30 min and then centrifuged at 14 000 r.p.m. for 30 min. The supernatant and the pellet were collected separately, boiled in tricine sample buffer and separated on a 16.5% tris–tricine gel. The gel was then stained with the InstantBlue™ protein stain (Expedeon Ltd, Cambridge, UK) and imaged using the UVP UVsolo *touch* (Analytik Jena, Germany).

## 4.8. Size-exclusion chromatography

Chromatography was performed on the AZURA® fast protein liquid chromatography system (KNAUER) in a buffer consisting of 50 mM HEPES-KOH pH 7.25, 150 mM KCl, 0.1 mM EDTA pH 8.0 and 10% glycerol. BSA (15 µM), CA (33 µM in the experiment shown in figure 9 and 15 µM in the experiment shown in electronic supplementary material, figure S4), MinE (13 µM), MinE-eYFP (13 µM) or eYFP (13 µM) were injected into a Superdex 200 column (GE Healthcare). The flow rate was set to 0.5 ml min$^{-1}$, and the run was performed at room temperature.

## 4.9. Image analysis and statistics

Cell length and mini-cells quantifications were performed using Fiji (https://fiji.sc/) [60].

## 4.10. Molecular modelling of MinE, MinE-eYFP, MinE-mCherry and MinE-iLOV structures

The NMR structure of the full-length, dimeric 6β-stranded form of *N. gonorrhoeae* MinE (PDB id: 2KXO, residues 1–87) and the partial X-ray crystal structure of the dimeric 4β-stranded *E. coli* MinE (PDB id: 3R9 J, residues 12–88) were used as templates for molecular modelling. We applied a two-steps approach to generate the complete model structures of the 6β and 4β *E. coli* MinE and MinE-eYFP dimers. First, the 3β and 2β MinE monomer structures were modelled, and then the 6β and 4β MinE dimer structures were obtained by using the following protocol: the MODELLER 9.21 software [43] was used to model the monomeric *E. coli* MinE 3β-stranded structure using PDB id: 2KXO as a template and the monomeric *E. coli* MinE 2β-stranded structure using PDB id: 3R9 J as a template. The 3β and 2β model structures were then aligned to their MinE dimer templates to generate the 6β- and 4β-stranded full MinE dimer structures. The 3β and 2β monomer structures of MinE-eYFP were modelled using PDB id: 1OXD, 2KXO and 3R9 J as templates. Then, the generated 3β and 2β monomer MinE-eYFP structures were aligned to their MinE dimer templates to obtain the 6β and 4β MinE-eYFP dimer structures.

The same approach was also applied to model the 3β and 2β monomer structures of MinE-iLOV (using as templates the following PDB ids: 4EES, 2KXO and 3R9 J) and MinE-mCherry (using as templates the following PDB ids: 1OXD, 2KXO and 3R9 J). The generated 3β and 2β monomer MinE-iLOV and MinE-mCherry structures were then aligned to the MinE dimer templates to obtain the 6β and 4β MinE fusion dimer structures.

The secondary structure elements of the 6β and 4β model structures of *E. coli* MinE were analysed using the STRIDE web-server [61], and surfaces important for MinE function, namely the MTS, the MinD-binding interface, the MinE dimerization interfaces 1 and 2, and the MinE polymerization interface, were assigned on the secondary structure elements.

## 4.11. CG REMD simulations

We used the UNRES web-server [46] to run five independent CG REMD simulations of 1 000 000 steps each for the 3β (monomer) and 4β (dimer) model MinE-eYFP structures. For the 3β (monomer) and the 4β (dimer) model structures of MinE-iLOV and MinE-mCherry, we conducted three independent CG REMD simulations of 1 000 000 steps for each run. The first cluster structures output of each REMD simulation were recorded. Diffusion accessibility calculations were conducted for the structures of the monomer and dimer 6β and 4β MinE and MinE fusions on a web-based platform [62].

Data accessibility. Raw data can be requested from the corresponding author.

Authors' contribution. N.P. and B.D.V. conceived the experiments. N.P. performed all experiments except fluorescence microscopy, which was performed by E.B.A. M.A.Ö. performed all computational analyses. All authors discussed, analysed and interpreted the data. B.D.V. wrote the manuscript with input from N.P. and M.A.Ö.

Competing interests. The authors declare they have no competing interests.

Funding. This study was funded by the DFG (grant no. VE776/2-1 to B.D.V.), by the BMBF (grant no. 031L0079 to B.D.V.) and by the Excellence Initiative of the German Federal and State Governments BIOSS (Centre for Biological Signalling Studies; EXC-294).

Acknowledgement. We thank João Nuno de Sousa Machado for performing size-exclusion chromatography and Sonja-Verena Albers for critical reading of this manuscript.

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
