## [Reviewer comments · Open Biology]

Review History

RSOB-20-0010.R0 (Original submission)

Review form: Reviewer 1

Recommendation

Reject – article is not of sufficient interest (we will consider a transfer to another journal)

Do you have any ethical concerns with this paper?

No

Comments to the Author

The manuscript 'C-terminal eYFP fusion impairs *Escherichia coli* MinE function' by Palanisamy et. al. focuses on the biochemical evidence for the abnormal function of MinE-eYFP. Although the biochemical characterization is done in detail, there are concerns about the novelty and originality. It is known for a long time that a fluorescent protein fusion may interfere with the physiological function of the target protein and it should be used with caution. In addition, as mentioned several times in the manuscript, partial function of the fluorescent protein fused with MinC, MinD, or MinE was reported or discussed in the past. More importantly, the data are insufficient to systematically demonstrate the degree of impairments in studying Min oscillation. Since almost all current technologies require genetic engineering or chemical modification on the protein to allow time-lapse, live-cell imaging and most cell division proteins are sensitive to these manipulations, most colleagues have tried hard to reduce the abnormality when using the

fluorescent protein fusion constructs. That is, a lot of efforts have been put in to ensure the best physiological outcomes under given circumstances. The detailed characterization of the defective fusion proteins was often not included in the published papers in the old days due to length limitation of the journal.

Review form: Reviewer 2

Recommendation

Accept with minor revision (please list in comments)

Do you have any ethical concerns with this paper?

No

Comments to the Author

Title: C-terminal eYFP fusion impairs Escherichia Coli MinE function

In this study the authors address the question to which extent a C-terminal eYFP fusion of E. coli MinE impairs its function. This question goes back to contradictory reports in literature about MinE function. In 1997, a study about MinE concluded that a MinE-GFP fusion is fully functional. Since then several reports have indicated that the functionality of a C-terminal MinE fusion is impaired.

The strength of this paper lies in the fact that the authors not only addressed the question about the functionality, but also investigated which parts and functionalities of MinE are affected by the C-terminal fusion of eYFP. Using different *in vivo* and *in vitro* experiments the authors showed that MinE-eYFP is only partially functional. It still allows the characteristic oscillatory behavior of the Min system in a certain concentration range but tends to aggregate in E. coli cells. Furthermore, MinE-eYFP is impaired in its ability to remove MinC from MinD, to stimulate the ATP hydrolysis in MinD and to attached to the membrane as shown by co-sedimentation assays.

In a second part, the study furthers addresses the question as to which part of MinE is affected by the C-terminal fusion using coarse-grained replica-exchange molecular dynamics simulations, analyzing the accessibility of different structural elements in MinE and MinE-eYFP. The simulations indicated that neither MinE-eYFP dimerization nor MinD binding is compromised, while a less accessible MTS points towards weaker membrane interaction. The simulations also predicted that R21 a crucial residue for ATPase stimulation is less accessible in MinE-eYFP offering an explanation for the observed decrease in ATPase activity in presence of MinE-eYFP compared to MinE. Unfortunately, the structures in Figure 6 are too small and the relevant details, which are crucial for understanding the argumentation are hardly recognizable. It would also be helpful to make detailed view of the position of R21 in the structure of MinE.

In general, the experiments are well designed and executed and the methods nicely complement each other. The study would be broader if the authors were able to assess whether other C-terminal MinE fusions (e.g. MinE-eGFP/MinE-RFP) show the same behavior as MinE-eYFP in E. coli MG1655 Δ minB. This is especially important regarding the report about MinE-GFP being fully functional, while the authors only investigate MinE-eYFP. The suggested experiments consolidate these data and might answer the question whether a C-terminal MinE fusion generally impairs MinE function.

Minor:

1. In Figure S1A the authors show that eyfp expressed in E. coli tends to form aggregates on its own. A MinE fusion with a protein (e.g. GST) that is less problematic seems to be helpful to discriminate the contribution of eYFP and MinE to the aggregation process.

2. The authors state that the oscillation behavior in the case of MinCDE-eYFP transfection seems to be normal in the case of low arabinose induction. Does the oscillation frequency change in comparison to wildtype?
3. Molecular weight labelling is missing on the SDS PAGES in Figure 5A.
4. In figure 6 the structures are too small and the structural features are hardly visible due to the surface overlay. It would be helpful to show the structures separately and indicate R21.
5. The SEC experiment (Figure 8) needs a further control to exclude the dimerization of eYFP on its own: SEC profile of eYFP.
6. Wrong figure reference: In the second last results paragraph: Figure 8C should be Figure 7C

Decision letter (RSOB-20-0010.R0)

11-Feb-2020

Dear Professor Di Ventura,

We are writing to inform you that the Editor has reached a decision on your manuscript RSOB-20-0010 entitled "C-terminal eYFP fusion impairs *Escherichia coli* MinE function", submitted to Open Biology.

As you will see from the reviewers' comments below, there are a number of criticisms that prevent us from accepting your manuscript at this stage. The reviewers suggest, however, that a revised version could be acceptable, if you are able to address their concerns. If you think that you can deal satisfactorily with the reviewer's suggestions, we would be pleased to consider a revised manuscript.

The revision will be re-reviewed, where possible, by the original referees. As such, please submit the revised version of your manuscript within four weeks. If you do not think you will be able to meet this date please let us know immediately.

When submitting your revised manuscript, please respond to the comments made by the referee(s) and upload a file "Response to Referees" in "Section 6 - File Upload". You can use this to document any changes you make to the original manuscript. In order to expedite the processing of the revised manuscript, please be as specific as possible in your response to the referee(s).

Sincerely,
The Open Biology Team
<mailto:openbiology@royalsociety.org>

Reviewer(s)' Comments to Author(s):

Referee: 1

Comments to the Author(s)

The manuscript 'C-terminal eYFP fusion impairs Escherichia coli MinE function' by Palanisamy et. al. focuses on the biochemical evidence for the abnormal function of MinE-eYFP. Although the biochemical characterization is done in detail, there are concerns about the novelty and originality. It is known for a long time that a fluorescent protein fusion may interfere with the physiological function of the target protein and it should be used with caution. In addition, as mentioned several times in the manuscript, partial function of the fluorescent protein fused with MinC, MinD, or MinE was reported or discussed in the past. More importantly, the data are insufficient to systematically demonstrate the degree of impairments in studying Min oscillation. Since almost all current technologies require genetic engineering or chemical modification on the protein to allow time-lapse, live-cell imaging and most cell division proteins are sensitive to these manipulations, most colleagues have tried hard to reduce the abnormality when using the fluorescent protein fusion constructs. That is, a lot of efforts have been put in to ensure the best physiological outcomes under given circumstances. The detailed characterization of the defective fusion proteins was often not included in the published papers in the old days due to length limitation of the journal.

Referee: 2

Comments to the Author(s)

Title: C-terminal eYFP fusion impairs Escherichia Coli MinE function

In this study the authors address the question to which extent a C-terminal eYFP fusion of E. coli MinE impairs its function. This question goes back to contradictory reports in literature about MinE function. In 1997, a study about MinE concluded that a MinE-GFP fusion is fully functional. Since then several reports have indicated that the functionality of a C-terminal MinE fusion is impaired.

The strength of this paper lies in the fact that the authors not only addressed the question about the functionality, but also investigated which parts and functionalities of MinE are affected by the C-terminal fusion of eYFP. Using different in vivo and in vitro experiments the authors showed that MinE-eYFP is only partially functional. It still allows the characteristic oscillatory behavior of the Min system in a certain concentration range but tends to aggregate in E. coli cells. Furthermore, MinE-eYFP is impaired in its ability to remove MinC from MinD, to stimulate the ATP hydrolysis in MinD and to attached to the membrane as shown by co-sedimentation assays.

In a second part, the study furthers addresses the question as to which part of MinE is affected by the C-terminal fusion using coarse-grained replica-exchange molecular dynamics simulations, analyzing the accessibility of different structural elements in MinE and MinE-eYFP. The simulations indicated that neither MinE-eYFP dimerization nor MinD binding is compromised, while a less accessible MTS points towards weaker membrane interaction. The simulations also predicted that R21 a crucial residue for ATPase stimulation is less accessible in MinE-eYFP offering an explanation for the observed decrease in ATPase activity in presence of MinE-eYFP compared to MinE. Unfortunately, the structures in Figure 6 are too small and the relevant details, which are crucial for understanding the argumentation are hardly recognizable. It would also be helpful to make detailed view of the position of R21 in the structure of MinE.

In general, the experiments are well designed and executed and the methods nicely complement each other. The study would be broader if the authors were able to assess whether other C-terminal MinE fusions (e.g. MinE-eGFP/MinE-RFP) show the same behavior as MinE-eYFP in E.

coli MG1655 Δ minB. This is especially important regarding the report about MinE-GFP being fully functional, while the authors only investigate MinE-eYFP. The suggested experiments consolidate these data and might answer the question whether a C-terminal MinE fusion generally impairs MinE function.

Minor:

1. In Figure S1A the authors show that eyfp expressed in E. coli tends to form aggregates on its own. A MinE fusion with a protein (e.g. GST) that is less problematic seems to be helpful to discriminate the contribution of eYFP and MinE to the aggregation process.
2. The authors state that the oscillation behavior in the case of MinCDE-eYFP transfection seems to be normal in the case of low arabinose induction. Does the oscillation frequency change in comparison to wildtype?
3. Molecular weight labelling is missing on the SDS PAGES in Figure 5A.
4. In figure 6 the structures are too small and the structural features are hardly visible due to the surface overlay. It would be helpful to show the structures separately and indicate R21.
5. The SEC experiment (Figure 8) needs a further control to exclude the dimerization of eYFP on its own: SEC profile of eYFP.
6. Wrong figure reference: In the second last results paragraph: Figure 8C should be Figure 7C

Author's Response to Decision Letter for (RSOB-20-0010.R0)

See Appendix A.

RSOB-20-0010.R1 (Revision)

Review form: Reviewer 2

Recommendation

Accept as is

Do you have any ethical concerns with this paper?

No

Comments to the Author

The authors addressed all points that I raised and I am pleased with the result. The simulations of the MinE-mCherry and MinE-iLOV are a nice addition to the manuscript. I especially like the restructured figures 6 and 7 which to me are much clearer now and makes it easier to understand the modelling.

I do agree with the authors that performing all the in vitro experiments with MinE-mCherry and MinE-iLOV would be beyond the scope of the manuscript, however, I would have liked to see how these fluorescent MinE variants behave in E. coli MG1655 Δ minB, to see their respective effect in vivo. Nevertheless, I recommend this paper for publication, as the experiments are not crucial for the core message of the paper.

Decision letter (RSOB-20-0010.R1)

29-Apr-2020

Dear Professor Di Ventura,

We are pleased to inform you that your manuscript entitled "C-terminal eYFP fusion impairs *Escherichia coli* MinE function" has been accepted by the Editor for publication in Open Biology.

If applicable, please find the referee comments below. No further changes are recommended.

Article processing charge

Please note that the article processing charge is immediately payable. A separate email will be sent out shortly to confirm the charge due. The preferred payment method is by credit card; however, other payment options are available.

Sincerely,

The Open Biology Team

mailto: openbiology@royalsociety.org

Reviewer(s)' Comments to Author:

Referee: 2

Comments to the Author(s)

See attached file

Appendix A

Reviewer(s)' Comments to Author(s):

Referee: 1

Comments to the Author(s)

The manuscript 'C-terminal eYFP fusion impairs Escherichia coli MinE function' by Palanisamy et. al. focuses on the biochemical evidence for the abnormal function of MinE-eYFP. Although the biochemical characterization is done in detail, there are concerns about the novelty and originality. It is known for a long time that a fluorescent protein fusion may interfere with the physiological function of the target protein and it should be used with caution. In addition, as mentioned several times in the manuscript, partial function of the fluorescent protein fused with MinC, MinD, or MinE was reported or discussed in the past. More importantly, the data are insufficient to systematically demonstrate the degree of impairments in studying Min oscillation. Since almost all current technologies require genetic engineering or chemical modification on the protein to allow time-lapse, live-cell imaging and most cell division proteins are sensitive to these manipulations, most colleagues have tried hard to reduce the abnormality when using the fluorescent protein fusion constructs. That is, a lot of efforts have been put in to ensure the best physiological outcomes under given circumstances. The detailed characterization of the defective fusion proteins was often not included in the published papers in the old days due to length limitation of the journal.

We agree with Reviewer #1 that it has been known for a long time that fusing a protein of interest to GFP or its variants may impair its function. Indeed, the scope of this paper is not to reveal for the first time that fusion proteins might be dysfunctional. The scope of this paper is to offer a molecular understanding of what a C-terminal fusion may be doing specifically to *E. coli* MinE, resulting in its functional impairment. While we cannot judge if other scientists have come to our same conclusions and did not publish their results due to space limitations, we do believe that our results are novel and important as they also shade some light on the function of MinE, having therefore implications that go beyond the strict analysis of MinE-eYFP. Moreover, we would like to point out that we not only performed wet-lab experiments, but also used computational analyses to mechanistically understand what eYFP may be doing to MinE within the fusion. To the best of our knowledge, systematic analyses, driven by *in silico* studies, on the molecular details behind the functional impairment of fusions to fluorescent proteins are very rare.

Our manuscript can be considered a proof of concept study showing how *in vivo*, *in vitro* and especially *in silico* analyses can be combined to address the impact of a fluorescent tag on the functionality of the protein of interest. By iterating and complementing experimental and computational analyses, we were able to get detailed and mechanistic understanding of the MinE-eYFP fusion protein. We believe

that our study will inspire other researchers, resulting in more studies in the future combining experimental and computational approaches to analyze fusion proteins, likely leading to generally applicable solutions to the protein fusion problem.

Beyond this, our paper offers a relatively comprehensive set of assays to assess the functions and properties of MinE, and includes data and information that have been originally presented in different scattered publications, making it now easier to find them. Our paper also contains novel datasets on untagged MinE. In Figure 8, we show that a truncated MinE deprived of the MTS can trigger the release of MinD from the liposomes as well as wild-type MinE. While this information could be *deduced* from the publication by Kretschmer, Zieske and Schille (PLoS ONE, 2017), where the authors show that truncated MinE (which they call MinE $\Delta(2-12)$, corresponding to what we call MinE¹³⁻⁸⁸) can stimulate MinD ATPase activity using *in vitro* ATPase assays, we believe it is valuable to confirm these results using a different assay (liposome co-sedimentation with MinD).

Finally, regarding the fact that our data are “insufficient to systematically demonstrate the degree of impairments in studying Min oscillation”: we agree with Reviewer #1 that it would be useful to compare the frequency of the oscillations obtained with MinE and MinE-eYFP. We admit that we had not thought of doing this before, because studying Min oscillations was not the focus of our paper, which rather revolved around understanding the mechanism leading to the impairment of the fusion protein. The right way of measuring the impact of eYFP on the oscillations would be *in vitro*, using pattern formation assays, which are not within our expertise. *In vivo* measurements of the frequency of the oscillations is not the truly appropriate experiment, since, when expressing inside cells the two constructs – one with MinE, one with MinE-eYFP – it is not possible to exclude that the observed effects on the frequency be due to differences in protein levels rather than functionality. This is the same reasoning that prompted us to move away from the *in vivo* complementation assay and perform *in vitro* studies instead (we mention this in the first sentence of the paragraph entitled “*MinE-eYFP is prone to aggregation*”). Interestingly, in their EMBO Journal paper in 2001, Hale, Meinhardt and de Boer observed slower oscillations for MinE-GFP than GFP-MinD but conclude that it is not possible to discern if this be due to MinE-GFP being expressed at lower levels or being functionally impaired compared to MinE. We now demonstrate with this paper that MinE-eYFP is functionally impaired.

Despite these counter arguments, during the revision time we nonetheless tried to measure the oscillation frequency *in vivo*, by fluorescently labeling MinD with eYFP and expressing it with either MinE or MinE-eYFP*, where the star indicates a mutant non-fluorescent version of eYFP. We cloned two bi-cistronic constructs, one co-expressing eYFP-MinD with MinE and one co-expressing eYFP-MinD and MinE-eYFP^{Y145W}, where Y145W is a mutation that was shown to almost entirely eliminate the fluorescence of MinE (Ganesan et al., PNAS, 2006; 103(11): 4089–4094.). We designed the experiment in a way that (a) we would leave largely untouched the

protein we studied *in vitro* (namely MinE-eYFP; we reckon that the single point mutation Y145W does not cause major structural re-arrangements to the protein) and (b) we would measure the oscillations by looking at the same protein in both cases, namely MinD. We decided not to use a different fluorescent protein, compatible with eYFP, such as mCherry, on MinD, because we lately made some (unpublished) observations that mCherry-MinD may behave strangely and differently from eYFP-MinD, which we used in the past and we trust (Di Ventura and Sourjik, Mol Syst Biol, 2011). We also, obviously, did not want to swap the fluorescent protein on MinE, since MinE-eYFP is the protein we used in all our *in vitro* assays. Unfortunately, when looking at the samples at the microscope, we discovered that eYFP^{Y145W} is still very fluorescent, which impedes the quantification of the oscillations, since we obtain a signal from both MinD and MinE at the same time. Looking in the literature for an explanation, we came across the paper by Murakoski and colleagues, in which they claim that the A206K mutation, introduced to monomerize eYFP, increases the brightness of the protein and counterbalances the effect of the Y145W mutation (Brain Cell Biol, 2008; 36:31-42). We do have the A206K mutation in our eYFP, thus the originally puzzling results are explained. While it would be in theory possible to test other mutations that could abolish the fluorescence of eYFP^{A206K}, we deem such efforts unjustified given the reason detailed at the beginning of the inherent inappropriateness of such *in vivo* measurements. We have now added a sentence in the Discussion about the impact of the fluorescent protein on the oscillation frequency.

Referee: 2

Comments to the Author(s)

Title: C-terminal eYFP fusion impairs Escherichia Coli MinE function

In this study the authors address the question to which extent a C-terminal eYFP fusion of E. coli MinE impairs its function. This question goes back to contradictory reports in literature about MinE function. In 1997, a study about MinE concluded that a MinE-GFP fusion is fully functional. Since then several reports have indicated that the functionality of a C-terminal MinE fusion is impaired.

The strength of this paper lies in the fact that the authors not only addressed the question about the functionality, but also investigated which parts and functionalities of MinE are affected by the C-terminal fusion of eYFP. Using different *in vivo* and *in vitro* experiments the authors showed that MinE-eYFP is only partially functional. It still allows the characteristic oscillatory behavior of the Min system in a certain concentration range but tends to aggregate in E. coli cells. Furthermore, MinE-eYFP is impaired in its ability to remove MinC from MinD, to stimulate the ATP hydrolysis in MinD and to attached to the membrane as shown by co-sedimentation assays.

In a second part, the study further addresses the question as to which part of MinE is affected by the C-terminal fusion using coarse-grained replica-exchange molecular dynamics simulations, analyzing the accessibility of different structural elements in MinE and MinE-eYFP. The simulations indicated that neither MinE-eYFP dimerization nor MinD binding is compromised, while a less accessible MTS points towards weaker membrane interaction. The simulations also predicted that R21 a crucial residue for ATPase stimulation is less accessible in MinE-eYFP offering an explanation for the observed decrease in ATPase activity in presence of MinE-eYFP compared to MinE. Unfortunately, the structures in Figure 6 are too small and the relevant details, which are crucial for understanding the argumentation are hardly recognizable. It would also be helpful to make detailed view of the position of R21 in the structure of MinE.

In general, the experiments are well designed and executed and the methods nicely complement each other. The study would be broader if the authors were able to assess whether other C-terminal MinE fusions (e.g. MinE-eGFP/MinE-RFP) show the same behavior as MinE-eYFP in *E. coli* MG1655 Δ minB. This is especially important regarding the report about MinE-GFP being fully functional, while the authors only investigate MinE-eYFP. The suggested experiments consolidate these data and might answer the question whether a C-terminal MinE fusion generally impairs MinE function.

We thank Reviewer #2 for recognizing the strength of our paper and for finding our experiments “well designed and executed”, with “methods that nicely complement each other”. We are grateful for the many useful comments and for the suggested additional experiments/controls and simulations, which we performed in full. To reply to the reviewer’s points, we have edited the main text and the figures. Specifically, we split old Figure 6 into two figures, so that the structures could be larger and details better visible, added a panel to existing supplementary figure S1, and added supplementary figures 3 and 4. To be able to generalize our findings, we performed the same computational analyses with two additional fluorescent proteins, namely MinE-mCherry and MinE-iLOV (data shown in new supplementary figure S5). We selected mCherry because it is not a variant of GFP obtained by mutations (we do expect little differences between GFP and eYFP). We selected iLOV because, being a relatively small protein (~10 kDa), we hoped it would interfere less with MinE structural features and that it could represent a solution to the problem of MinE tagging. However, the simulations show that, despite leaving certain features unaffected (e.g. the accessibility of the MTS is better with these two proteins compared to eYFP), both mCherry and iLOV likely impact other aspects of MinE, such as the accessibility of R21 and MinE dimerization. Surely, testing these predictions experimentally would be nice. However, we hope that the reviewer understands that it would mean truly a large amount of work to clone, purify and perform the *in vitro* assays once again. We decided to show the results of the new

simulations in the supplement, since these remain only predictions and were not experimentally tested. Nonetheless, we believe it is fair to generalize the conclusions and speculate that same results would hold true for other fluorescent proteins.

Below we reply to the minor points raised.

Minor:

1. In Figure S1A the authors show that *eyfp* expressed in *E. coli* tends to form aggregates on its own. A MinE fusion with a protein (e.g. GST) that is less problematic seems to be helpful to discriminate the contribution of eYFP and MinE to the aggregation process.

Following the reviewer's suggestion, we cloned MinE-GST and tested its expression. We found that the fusion protein goes almost entirely to the pellet, like MinE-eYFP (new Suppl. Fig. S1). We however read in the literature that GST dimerizes and therefore feared that it may be triggering aggregation of MinE since MinE also dimerizes. To clarify this issue, we additionally cloned MinE-MBP, since MBP is monomeric and is often used as a solubility tag in protein expression and purification experiments. We found that MinE-MBP behaves more similarly to untagged MinE, with bands both in the supernatant and the pellet, albeit the most abundant protein band is found in the pellet. While these results could indicate that indeed eYFP brings MinE-eYFP to the pellet due to its intrinsic aggregation propensity, we rather disfavor this explanation since we show in new Suppl. Fig. S4 that the eYFP we use (eYFP^{A206K}) is monomeric. Suppl. Fig. S1 also shows that eYFP^{A206K} alone is mostly soluble. All in all, we think that MinE-eYFP is found in the pellet due to the properties of the fusion and not due to eYFP *per se*.

2. The authors state that the oscillation behavior in the case of MinCDE-eYFP transfection seems to be normal in the case of low arabinose induction. Does the oscillation frequency change in comparison to wildtype?

We thank the reviewer, as well as Reviewer #1, for prompting us to think in terms of the oscillation frequency. As written in our response to Reviewer #1, indeed we had not previously thought of quantifying the frequency of the oscillations, because we were focusing on understanding the mechanism by which eYFP impairs MinE. We think that the right way of measuring the impact of eYFP on the oscillation frequency would be by *in vitro* pattern formation, since this is the only way to ensure that MinE and MinE-eYFP be at the same concentration. Such *in vitro* pattern formation studies are not within our expertise. When expressing the constructs inside cells, we cannot exclude that the observed effect on the oscillations be due to differences in protein levels rather than functionality. This is the same reason why we decided to perform *in vitro* assays moving away from the *in vivo* complementation assay (we mention this at the beginning of the paragraph entitled: "*MinE-eYFP is prone to aggregation*"). Actually, this conclusion was also reached by Hale, Meinhardt and de Boer. In 2001,

in their EMBO Journal paper, they observed slower oscillations for MinE-GFP compared to those of GFP-MinD but conclude that it is not possible to discern if this be due to MinE-GFP being expressed at lower levels compared to MinE or being functionally impaired. Our paper now finally clarifies that MinE-eYFP is functionally impaired.

Said that, we actually nonetheless tried, during the revision time, to measure *in vivo* the frequency of MinD oscillations, by cloning two constructs: eYFP-MinD/MinE and eYFP-MinD/MinE-eYFP*, where the star means that we introduced a mutation to the eYFP fused to MinE to render it non-fluorescent. Specifically, we used the previously reported Y145W mutation (Ganesan et al., PNAS, 2006; 103(11): 4089–4094.). We thought this was a good strategy, because we would be following the very same protein (MinD) and not, as done previously, two different ones (namely MinD and MinE, by employing GFP-MinD and MinE-GFP). See also our reply to Reviewer #1 for additional information on the logic behind this experiment (why we selected eYFP and not mCherry, etc). However, sadly, we discovered that the Y145W mutation, in the context of eYFP^{A206K}, does not eliminate fluorescence (finding that is corroborated by the paper by Murakoski and colleagues (Brain Cell Biol, 2008; 36:31-42), which we found afterwards, when looking for a justification of our data in the literature). Since correcting this issue would take us long – we would have to either eliminate the A206K mutation, introduce a different one to monomerize eYFP, re-clone and re-perform the microscopy experiments or try other mutations to eliminate fluorescence – we have finally decided against performing this experiment due to the reason mentioned at the beginning of the inherent inappropriateness of such *in vivo* measurements. We have now added a sentence in the Discussion about the impact of the fluorescent protein on the oscillation frequency.

3. Molecular weight labelling is missing on the SDS PAGES in Figure 5A.

We thank the reviewer for spotting this. We now added the labeling.

4. In figure 6 the structures are too small and the structural features are hardly visible due to the surface overlay. It would be helpful to show the structures separately and indicate R21.

We split old Figure 6 in two, so that the structures could be drawn larger. We also indicate R21 on the structure. We additionally added in the new main Figure 7 examples of input and output of the simulations. We hope this aids in the interpretation of the results of the computational analysis.

5. The SEC experiment (Figure 8) needs a further control to exclude the dimerization of eYFP on its own: SEC profile of eYFP.

We performed SEC with EYFP^{A206K} alone and show that it is monomeric (new Supplementary Figure 4).

6. Wrong figure reference: In the second last results paragraph: Figure 8C should be Figure 7C

We thank the reviewer for spotting this mistake. We have corrected it and now refer to Figure 7C.